# Coordination of m^6^A mRNA Methylation and Gene Transcriptome in Sugarcane Response to Drought Stress

**DOI:** 10.3390/plants12213668

**Published:** 2023-10-24

**Authors:** Jinju Wei, Haibi Li, Yiyun Gui, Hui Zhou, Ronghua Zhang, Kai Zhu, Xihui Liu

**Affiliations:** 1Sugarcane Research Institute, Guangxi Academy of Agricultural Sciences, Nanning 530007, China; weijinju@126.com (J.W.); guiyiyun@gxaas.net (Y.G.); zhouhui@gxaas.net (H.Z.); zhr20056@163.com (R.Z.); 2Guangxi Key Laboratory of Sugarcane Genetic Improvement, Guangxi Academy of Agricultural Sciences, Nanning 530007, China; 3Guangxi South Subtropical Agricultural Science Research Institute, Guangxi Academy of Agricultural Sciences, Nanning 532415, China; lihaibi@gxaas.net

**Keywords:** N^6^-methyladenosine, transcriptome analysis, sugarcane, drought stress

## Abstract

The N^6^-methyladenosine (m^6^A) methylation of mRNA is involved in biological processes essential for plant growth. To explore the m^6^A modification of sugarcane and reveal its regulatory function, methylated RNA immunoprecipitation sequencing (MeRIP-seq) was used to construct the m^6^A map of sugarcane. In this study, m^6^A sites of sugarcane transcriptome were significantly enriched around the stop codon and within 3′-untranslated regions (3′UTR). Gene ontology (GO) analysis showed that the m^6^A modification genes are associated with metabolic biosynthesis. In addition, the m^6^A modification of drought-resistant transcript mRNA increased significantly under drought (DR) treatment, resulting in enhanced mRNA stability, which is involved in regulating sugarcane drought resistance. GO and Kyoto Encyclopedia of Genes and Genomes (KEGG) enrichment results showed that differentially methylated peak (DMP) modification of differentially expressed genes (DEGs) in DR were particularly associated with abscisic acid (ABA) biosynthesis. The upregulated genes were significantly enriched in the ABA metabolism, ethylene response, fatty acid metabolism, and negative regulation of the abscisic acid activation signaling pathway. These findings provide a basis and resource for sugarcane RNA epigenetic studies and further increase our knowledge of the functions of m^6^A modifications in RNA under abiotic stress.

## 1. Introduction

N^6^-methyl-adenosine (m^6^A) is a prevalent RNA epigenetic modification that affects mRNA in various ways and is critical in regulating plant development and evolution [1]. The m^6^A methyltransferase complex regulates early rice spikelet degeneration, thereby modulating sporogenesis [2]. Adenosine methylase deposits m^6^A on pri-miRNAs to regulate miRNA biosynthesis in Arabidopsis. The m^6^A modification in Arabidopsis is closely related to photosynthesis [3,4], indicating that m^6^A participates in regulating photosynthesis, affecting the plant circadian rhythm, flowering time, and polyadenylation site selection [5]. m^6^A demethylation regulates fruit ripening [6] and promotes the proliferation of root meristem tissue cells in rice and the formation of tiller buds, thereby enhancing photosynthetic efficiency and drought tolerance and increasing the yield of rice and potatoes [7]. The level of m^6^A modification in plants is dynamically regulated by methyltransferases and demethylases and participates in important biological processes.

Growing evidence indicates that m^6^A plays a role in regulating plant responses to various biotic and abiotic stressors [7]. The overall level of m^6^A in plants increases under stress from pathogenic microorganisms and other factors [3]. m^6^A also mediates mRNA decay, affecting the function of signaling and regulatory factors in biochemical reactions within plants [8]. In addition, m^6^A participates in regulating salt [1,9,10], cadmium [11], and cold stress [12] responses. Drought is the most significant abiotic stress factor affecting plant growth, development, and yield. Many studies have been conducted on m^6^A modifications induced by drought stress. In sea buckthorn, m^6^A modification can regulate transcription levels under drought stress [13]. Gene expression and protein relatedness decrease under drought stress and m^6^A modifications may negatively regulate the expression of genes related to wood development (cellulose, hemicellulose, and lignin) [14]. In summary, these studies demonstrated that m^6^A modifications play an important role in plant responses to stress; however, their regulatory mechanisms are complex and require further clarification.

The effect of drought stress on m^6^A methylation in sugarcane has not yet been reported. In this study, we used the methylated RNA immunoprecipitation (MeRIP) method to study the effect of drought stress on m^6^A methylation in sugarcane leaves of the variety ROC22. In the present study, we found that the overall m^6^A levels in sugarcane leaves significantly increased after drought stress. Therefore, we analyzed the m^6^A methylation profile of sugarcane leaves in response to drought stress using whole-genome m^6^A-seq and bioinformatic methods. In addition, we used RNA sequencing to analyze the response pattern and speculated on the functions of differentially expressed genes (DEGs) modified by m^6^A in the transcriptional profile of drought-stressed sugarcane leaves. Additionally, we validated the effectiveness of some target genes using qPCR (qPCR). This study represents the first comprehensive characterization of m^6^A modifications in the sugarcane transcriptome. Furthermore, we elucidated the potential roles and regulatory mechanisms of m^6^A-mediated sugarcane production in response to drought stress.

## 2. Results

### 2.1. m^6^A Methylation Is Abundant in Sugarcane Leaf mRNA

An m^6^A-targeted antibody was used to obtain the transcriptome-wide m^6^A map of sugarcane leaves, methylated RNA immunoprecipitation sequencing (MeRIP-seq) was performed, and a series of m^6^A-immunoprecipitation (IP) and matched input (non-IP control) libraries were constructed. We obtained almost 44 million reads per library (input and IP libraries) and detected 2826 m^6^A-enriched peaks in sugarcane leaf samples, which have an average length of 405 bp. The minimum peak length was 192 bp. Based on these results, we found that the sugarcane transcriptome contained 0.1–2.0 m^6^A peaks per 1000 nucleotides, which are similar to those obtained in mammals. Most of the genes contained one or two m^6^A peaks (Figure 1a).

m^6^A peak distribution was investigated based on gene annotation. Most reads of m^6^A peaks were located around the 3′UTRs (Figure 1b). Similar distribution patterns of m^6^A peaks have also been observed in Arabidopsis [15], sweet sorghum [9] and sea buckthorn [13]. The results of the locations of m^6^A on transcripts showed that m^6^A peaks were abundant around the stop codon (60.30%), followed by the 3′UTRs (21.76%), coding regions (14.93%), and the start codon (2.76%) (Figure 1c). The distribution pattern of m^6^A peaks in sugarcane was similar to that observed in maize [16] and Arabidopsis [17].

We also observed that m^6^A peaks were significantly enriched around the 3′UTRs (enrichment score > 90) using segment normalization of each gene portion (Figure 1d). The genome-wide density of m^6^A modification peaks showed that the m^6^A peaks had different distribution patterns across chromosomes (Figure 1e). The analysis revealed that the average density of chromosome 4C was the highest, and that of chromosome 6A was the lowest. The distribution patterns among these chromosomes suggested that m^6^A modifications may be related to chromatin status.

According to the analysis of the m^6^A consensus sequence, 15,387 (94.39%) m^6^A peaks had at least one RRACH motif in the CK group, and 10,873 (94.15%) m^6^A peaks in the DR group had at least one RRACH motif (Figure 1f). The same sequence motif observed in eukaryotic mRNA may be essential for m^6^A modifications.

Through analysis of m^6^A consensus sequences, we also observed that 2367 (83.76%) m^6^A peaks included at least one typical conserved sequence, RRACH, and 2564 (90.73%) m^6^A peaks included one typical conserved sequence, DRACH, in the CK group. The same sequence motifs have been observed in eukaryotic mRNA and may be crucial for m^6^A modification.

### 2.2. Characterization of m^6^A mRNA Methylation in Sugarcane

Previous studies demonstrated that m^6^A is crucial for plant development. To further analyze the function of m^6^A methylation in sugarcane, GO and KEGG enrichment analyses of genes modified by m^6^A under drought stress were performed. The result showed that most of these genes were involved in metabolic biosynthesis (Figure 2a,b). The genes were divided into three subgroups (Peak I: m^6^A peaks around the start codon; Peak II: m^6^A peaks around the stop codon; Peak III: m^6^A peaks around both the start and stop codons) based on the distribution of m^6^A peak values in different gene functional elements to determine the m^6^A distribution patterns of the genes related to their GO categories. The GO enrichment analysis of each subgroup showed that m^6^A-containing genes in Peak I were mainly enriched in photosynthesis-related processes (Figure 2c), those in Peak II were mainly enriched in protein synthesis and translation processes (Figure 2d), and those in Peak III exhibit high enrichment of organic compound biosynthetic and metabolic processes (Figure 2e). We identified a series of metabolism-related genes containing m^6^A peaks; for example, acetaldehyde dehydrogenase (*ScALDH7B4*) and *CLYI-1*, which contained m^6^A peaks near the stop codon and 3′UTR, playing important roles in the oxidation and detoxification of aldehydes and the carbohydrate metabolism, respectively. Glycerol-3-phosphate dehydrogenase A (*GAPA*) and citrate synthase (*CSY3*) play important roles in carbohydrate biosynthesis and also contain m^6^A peaks in the stop codon and 3′UTR (Figure 2f,g). These results indicated that m^6^A mRNA methylation may regulate the biosynthesis of plant metabolites.

### 2.3. The m^6^A Methylation Responses to Drought Stress in Sugarcane

The m^6^A methylomes in sugarcane leaves under CK and DR treatments were profiled to investigate changes in m^6^A methylation in sugarcane under drought stress. After comparing the distribution of m^6^A methylation in CK and DR, we identified 2826 and 6620 m^6^A methylation peaks in CK and DR, respectively. Venn diagram analysis revealed 1945 (25.90%) peaks in both CK and DR, with 881 peaks unique to CK and 4675 peaks unique to DR (Figure 3a). This indicates that m^6^A methylation increases under drought stress. Differential analysis showed that 2138 m^6^A peaks were differentially methylated peaks (DMPs) (*p* < 0.05, |log^2^FC| > 1), including 1373 upregulated and 765 downregulated DMPs (Figure 3b). Then, we detected 2208 transcripts that have DMPs in DR compared to CK. Among the 1373 upregulated DMPs related genes, including 150 up-regulated genes, 94 down-regulated genes, and 1168 unaltered genes; and among the 765 downregulated DMPs related genes, including 126 up-regulated genes, 32 down-regulated genes, and 637 unaltered genes, more than 40% of DMPs were located in the stop codon region (Figure 3c). Further analysis of the whole transcriptomes of CK and DR groups revealed that m^6^A modification affected gene expression levels (Figure 3d). GO enrichment analysis of genes containing DMPs was performed to explore the functional characteristics of DMPs in the context of their genetic location. GO enrichment analysis of the upregulated DMPs-containing genes showed that the biological processes of cell metabolism (GO:0044237), gene expression (GO:0010467), nitrogen compound metabolic processes (GO:0006807), and photosynthesis and light reaction (GO:0019684) were significantly enriched (Figure 3e). GO enrichment analysis of the downregulated DMPs-containing genes showed that the biological processes of protein ubiquitination (GO:0016567), nuclear-transcribed mRNA catabolic process (GO:0000956), and cell wall modification (GO:0042545) were significantly enriched. In addition, the KEGG analysis showed that the upregulated and downregulated DMPs contained genes that were mainly enriched in biological metabolic pathways (Figure 3g,h).

### 2.4. m^6^A Modification Regulates mRNA Abundance by Regulating the Stability of Drought-Tolerance Transcripts

A transcriptomic analysis of the sugarcane response to drought was performed to explore the effect of m^6^A methylation on the drought tolerance of sugarcane. Subsequently, the correlations between DMPs and differentially expressed genes (DEGs) was analyzed. A total of 10,994 DEGs, including 8436 upregulated and 2558 downregulated genes, were identified by comparing CK and DR treatments. There were 2138 DMPs in total (FDR < 0.05); 1373 upregulated DMPs were associated with 1326 upregulated DEGs, and 765 downregulated DMPs were associated with 769 downregulated DEGs (Figure 4a).

mRNA expression is affected by m^6^A modifications; therefore, there is a potential trend for co-regulation between gene expression in the transcriptome and the abundance of m^6^A peaks. We analyzed the DEGs identified in the transcriptome using the differentially identified peak-related genes in m^6^A. In total, 12,675 genes, including 364 differentially expressed genes, were identified (Figure 4b). Interestingly, when we analyzed transcripts with significant changes in m^6^A modification in sugarcane leaves, the mRNA abundance of some drought-resistant genes with significantly increased m^6^A modifications also increased significantly. For example, there were significant changes in m^6^A methylation around the stop codons of *OAT* (Sspon.01G0026110-2B), *NAC* (Sspon.01G0042440-1P), and *ALDH* (Sspon.02G0013900-3D), accompanied by a significant increase in mRNA abundance (Figure 4c).

Research has shown that increasing the proline content in plant cells can effectively enhance plant drought resistance; therefore, proline accumulation is closely related to plant drought stress. Proline biosynthesis in higher plants can be divided into two pathways, glutamate, and ornithine. *ScOAT* is a key enzyme in the ornithine pathway of proline synthesis and plays an important role in the response of sugarcane plants to drought stress [18]. *NAC* transcription factors not only play important roles in plant embryogenesis, flower patterning, secondary wall formation, leaf senescence, and lateral root development; they also participate in responses to biotic and abiotic stressors, such as pathogens, drought, salt, and cold stress [19,20]. With the assistance of *NAD*(P)+, *ALDH* converts a series of endogenous and exogenous aromatic and aliphatic aldehydes into their corresponding carboxylic acids, which are important enzymes for clearing reactive oxygen species in organisms and play an important role in maintaining material balance [21]. m^6^A modification of these key drought-resistant transcriptional transcripts increased mRNA stability and thus increased mRNA abundance (Figure 4d).

### 2.5. Changes in m^6^A Methylation-Related Genes in Response to Drought Stress in Sugarcane

To further explore the effect of drought stress on m^6^A methylation in sugarcane leaves, 12 drought-resistant genes in the CK and DR groups that were previously reported [18,21,22,23,24,25,26,27,28,29,30,31,32,33] were analyzed. Six genes were downregulated in the DMGs and upregulated in the DEGs (Table 1). Changes in methylation modification and up/downregulated gene expression of these drought-resistant genes may explain the phenotypic differences in sugarcane under drought stress.

We analyzed the role of the aldehyde dehydrogenase gene (*ALDH7*) with differential methylation and expression in the two sets of association analyses based on sup8 m^6^A+ transcriptome correlation analysis to further explore the relationship between m^6^A methylation and the transcriptome. Under drought and other stress conditions, plants may increase their abscisic acid (ABA) content and accumulate large amounts of free proline and arginine. Arginine and proline can be converted to putrescine (Spd) by PAQ1 and PAQ5, and putrescine can be converted to 4-aminobutanal by PAQ1. 4-aminobutanal can be converted into 4-aminobutyric acid by the action of the *ALDH7* enzyme, which is finally converted into succinate succinic acid. Arginine and proline can be converted into beta-amino propanal by the action of *PAQ1* and *AOC3*, and beta-amino propanal can be converted into beta-alanine by the action of *ALDH7*. Beta-alanine is further converted into pantothenic acid and pantothenic acid salts by PANC. In addition, the acetic acid salt in the cell can be converted into acetaldehyde by *ALDH7*, and acetaldehyde can be deaminated and converted into ethanol by *ALDH* (Figure 5). Previous studies have shown that butyric acid, pantothenic acid, and ethanol can improve drought resistance [18,25,34,35,36,37,38].

Aldehyde dehydrogenase plays a significant role as the core component of these three metabolic processes. It is noteworthy that the m^6^A level of the aldehyde dehydrogenase gene (*ALDH7*) is downregulated (down arrow), while the mRNA level is upregulated (up arrow). This indicates that the downregulation of m^6^A level may lead to an increase in mRNA expression level, thus resulting in the accumulation of a large amount of aldehyde dehydrogenase protein (*ALDHs*), which oxidizes aldehyde substances into the corresponding pantothenic acid, butyric acid, and ethanol, reduces lipid peroxidation, and ultimately participates in plant abiotic stress and developmental regulation.

### 2.6. Real-Time Quantitative PCR

In this study, nine DGEs were randomly selected to verify the expression of genes in the CK and DR sugarcane plants. Six upregulated (up arrow) and three downregulated (down arrow) genes involved in plant biotic metabolism, photosynthesis, and defense responses were selected. The upregulated and downregulated genes encoded aldehyde dehydrogenase, heat shock protein, ornithine aminotransferase, dehydration-responsive element-binding protein, fructose-bisphosphate aldolase cytoplasmic isozyme, pyrophosphate-energized vacuolar membrane proton pump, peptidylprolyl isomerase, and lactoylglutathione lyase. The results of the RT-PCR analysis are shown in Figure 6. Quantitative real-time analysis of the nine differential genes was compared to transcriptome data, and the results supported the m^6^A-seq quantification results.

## 3. Discussion

In the present study, we used m^6^A-seq technology to generate a transcriptome-wide RNA m^6^A modification profile in sugarcane leaves. We found that the distribution pattern of m^6^A in sugarcane is similar to that of other plants, such as Arabidopsis, sorghum, and rice [3,16,39], which are mainly enriched in the 3′UTR region and also have a similar RRm^6^ACH methylation motif in each transcript [3,40,41]. These results indicate that the distribution pattern of m^6^A methylation in plants is conserved. However, some differences were found between sugarcane, Arabidopsis, and sorghum, which may represent differences in m^6^A methylation levels and unique patterns between species or may be the result of different depths of m^6^A-seq in the studies. Previous studies with lower m^6^A resolutions may not be sufficient to identify weak m^6^A signals in most mRNA coding sequences, and earlier reports have suggested that their estimates of m^6^A sites may not be sufficient [40,42].

The deposition of m^6^A is positively correlated with mRNA levels, plays a regulatory role in plant gene expression, and regulates plant development and evolution [1]. Studies have shown that m^6^A deposition is negatively correlated with transcript abundance and interacts with DNA methylation; m^6^A demethylation regulates fruit ripening [6]. In the present study, drought stress increased the degree of m^6^A modification in crops and upregulated the expression of drought stress response genes, confirming that m^6^A modification of drought-stress-related transcripts is closely related to regulating plant responses to drought stress [9].

In the present study, we found that m^6^A modifications in sugarcane leaves changed significantly under drought stress. Some genes, such as *NAC002*, *HSP70*, *ALDH7B4*, *GLYI-11*, and *DREB1C*, showed decreased methylation levels but increased expression of drought-responsive genes, indicating that m^6^A may play an important role in the drought response. m^6^A modification can enhance the mRNA stability of some key drought-resistant genes, thereby increasing mRNA abundance, which is consistent with the conclusion of methylation changes in the salt tolerance of sorghum [9]. This study also suggests that m^6^A modification significantly changes plant development and abiotic stress. In addition, GO and KEGG analyses showed that the upregulated genes were significantly enriched in regulating abscisic acid metabolism and signaling pathways, such as glycine metabolism and activation of the abscisic acid signaling pathway (Figure 4d). For example, one of the members of the *Arabidopsis ALDH* family, *ALDH7B4*, and the key enzyme gene *OAT* in rice proline are important transcription factors in the ABA signaling pathway and play an important regulatory role in response to stress. In this study, compared to the CK samples, the mRNA m^6^A methylation levels of *OAT* and *ALDH7B4* in the DR samples increased four-fold, whereas the transcription levels of *OAT* and *ALDH7B4* in the DR samples were higher than those in the CK samples, responding to water shortage and enhancing the tolerance of sugarcane plants to drought stress. *PP2C50* is a serine/threonine residue protein phosphatase in the protein phosphatase family, which is involved in regulating ABA permeation and stress signaling pathways. It has hyper- and hypo-m^6^A peaks in the DR, which affect ABA expression levels. *SODA* is a superoxide dismutase family member involved in arginine and proline metabolism, whereas *cFBP* is involved in carbon metabolism. *SODA* and *cFBP* participate in the synthesis of abscisic acid metabolism. These results suggested that m^6^A methylation regulates the expression of ABA-related genes in sugarcane in response to drought stress.

These key drought resistance genes play a positive role in regulating the response of sugarcane plants to drought stress. The expression of drought-responsive genes is the basis for plant drought resistance [43]. Most genes induced via drought stress are closely related to cellular regulation, protein metabolism, signal transduction, and transcription [44]. Although only a few key drought-resistant transcripts can be accurately detected to directly regulate mRNA abundance through m^6^A modification, the regulatory effects of m^6^A modification on mRNA are complex, and most m^6^A modifications may regulate mRNA by affecting RNA metabolism. It can be speculated that more m^6^A-modified critical drought-resistant transcripts will provide more complex water stress regulatory pathways, such as RNA splicing, RNA export, 3′UTR processing, and translation, thus making plants more drought-resistant.

Under water stress, sugarcane can adjust the expression of downstream related genes and directly act as functional proteins to protect plant cells from the impact of water stress, ultimately enabling plants to exhibit adaptability or resistance to water stress [45,46,47,48,49]. Studies have shown that ABA not only regulates the entire development process of plants but also plays a bridging role in different environmental signals. ABA is a key factor for plant cells in adapting to drought and plays an important role in plant stress responses by initiating drought response mechanisms [50,51]. At the same time, it has been found that in the drought response process, ABA can not only change the transcription levels of plant cells but also affect other important basic metabolic processes of plants, including plant lipid and carbon metabolism, which also implies the central role of ABA in plant responses to drought [17]. In this study, *ALDH7B4*, a gene related to ABA metabolism and synthesis, showed a decrease in m^6^A methylation levels, indicating that m^6^A modification of the ABA metabolic synthesis pathway is a stress response of plant drought resistance caused by environmental drought stress. Upregulation of *ALDH7B4* transcription promotes the ability of the cell to regulate permeability under adverse conditions. For plant drought resistance, regulating drought-resistant genes is a quantitative trait, an alternative to relying on a single gene for a decisive role. Therefore, the m^6^A modification of drought-resistant transcripts may provide more signaling pathways for regulating plants under water stress.

## 4. Materials and Methods

### 4.1. Plant Materials and RNA Isolation

The sugarcane variety ROC22 (*Saccharum* spp. hybrid) was provided by the Sugarcane Research Institute, Guangxi Academy of Agricultural Sciences, Nanning, China (108°19′ E, 22°49′ N). Sugarcane seed pieces with single buds were cultivated using a substrate, and healthy cane seedlings were selected and planted in 10 L pots 35 cm × 30 cm (height × diameter) in size. Drought stress was applied to sugarcane after 3 months of growth. Sugarcane was naturally drought-stressed in a greenhouse with controlled light, temperature, and humidity conditions (16 h dark, 8 h light; 29–31 °C under day and dark, 60–70% relative humidity). Soil water content in the drought resistance (DR) group was 7–9% after 10 days of drought treatment, whereas it was maintained at 25–28.5% in the control check (CK) group. The +1 leaves of CK and DR plants were randomly sampled, rapidly frozen in liquid nitrogen, and stored at −80 °C for RNA extraction and sequencing.

### 4.2. RNA Isolation and Library Construction

Total RNA was extracted from leaves using the TRIzol (Qiagen, Beijing, China) method, and the quality of RNA samples was determined using a Nanodrop microspectrophotometer and gel electrophoresis. Polyadenylated RNA was extracted and fragmented to 100 nt using a fragmentation reaction system. The fragment RNA and m^6^A-specific antibody were incubated in IP buffer at 4 °C for 2 h for immunoprecipitation. The mixture was eluted with IP buffer, the precipitated RNA was recovered, and immunoprecipitation mRNA (IP) and pre-immunoprecipitation mRNA (Input) libraries were constructed for each sample using an Illumina Ribo-Zero Gold kit (MRZG12324, Illumina, San Diego, CA, USA). We performed m^6^A-seq and input RNA-seq with NovaSeq 6000 using a paired-end strategy.

### 4.3. Processing of Raw Data

The reads obtained from the sequencing machines included raw reads containing adapters or low-quality bases which would affect the following assembly and analysis. Raw reads would be processed to get high quality clean reads according to four stringent filtering standards use fastp software (version: 0.20.0) [40]. Clean reads were mapped to the sugarcane (*Saccharum spontaneum* L.) reference genome (https://www.life.illinois.edu/ming/downloads/Spontaneum_genome/ accessed on 9 December 2021) using HISAT2 software (version:2.1.0). We used the R package (exomePeak2) (version:1.0.0) to identify m^6^A peaks (*p*-value < 0.05); then, the distribution of m^6^A in the different parts of the gene and transcripts was estimated.

Motif analysis was performed using MEME Suite (http://meme-suite.org/ accessed on 11 December 2021) to obtain the precise location and motif regions of the m^6^A peaks. The Integrated Genome Browser was used for visual analysis of m^6^A peaks [52]. All m^6^A modification sites were assigned to different transcriptional regions, including transcription start sites (TSS), 5′UTR, coding sequences (CDS), 3′UTR, and introns. Gene expression levels were calculated using the reads per kilobase per million mapped reads (RPKM) method. The m^6^A differential site algorithm was used to determine the differential m^6^A peaks between CK and DR sugarcane leaves [53], with a *p*-value of <0.05. Gene ontology (GO) and Kyoto Encyclopedia of Genes and Genomes (KEGG) enrichment analyses were used to identify the biological processes of differentially modified genes.

### 4.4. Quantitative Real-Time PCR (qRT-PCR) Validation

Previous studies have shown that quantitative real-time PCR (qRT-PCR) can be used to validate RNA-seq results. We analyzed the expression patterns of the selected genes. RNA extraction, cDNA synthesis, and qRT-PCR were performed as previously described. Transcription levels of the selected genes were quantified using the cycle threshold 2^−ΔΔCt^ method. The sugarcane *GAPDH* gene (GenBank accession number: EF189713) was used to normalize the expression values. The primers used for qRT-PCR are listed in Appendix A. Experiments were conducted using three biological replicates.

## 5. Conclusions

This study reports the effects of drought stress on m^6^A mRNA methylation and the expression of related genes in sugarcane leaves. Using MERIP-seq, we constructed a transcriptome-wide m^6^A modification map of sugarcane and demonstrated the role of m^6^A modification in the response to drought stress. Our results suggest that the number and extent of m^6^A modifications in drought-resistance-related mRNA may be important parameters in assessing the drought tolerance of crops.

## Figures and Tables

**Figure 1 plants-12-03668-f001:**
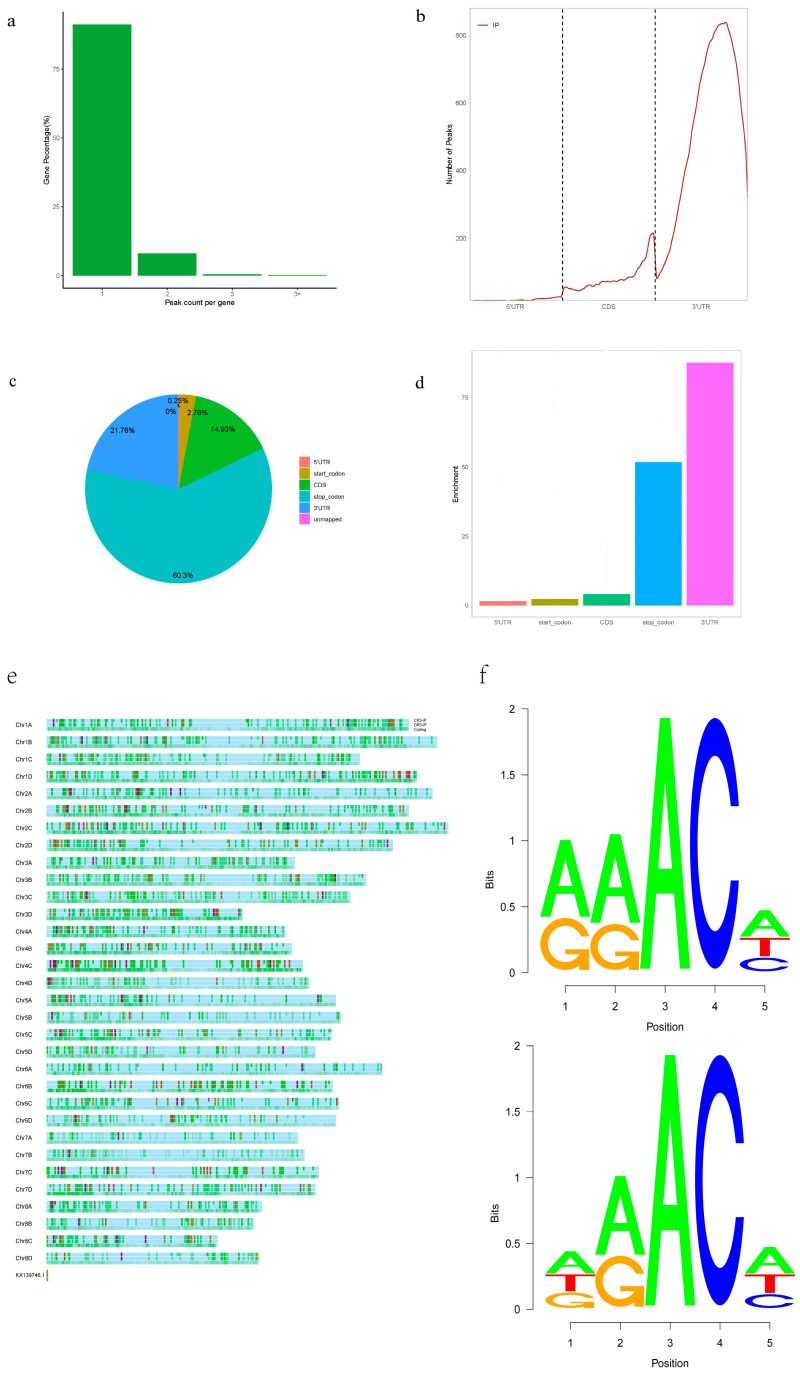
Overview of m^6^A methylome in sugarcane. (**a**) Number of genes with different m^6^A peak numbers. (**b**) Accumulation of m^6^A-IP and input reads of transcripts. Each transcript contained three parts: 5ʹUTRs, CDS, and 3′UTRs. (**c**) The m^6^A peak distribution in different genic regions. (**d**) The m^6^A peak distribution along a metagene. Enrichment scores are calculated as (n/N)/p; n, number of peaks belonging to each category; N, number of total peaks; p, proportion of each category within the genome by length. (**e**) Distribution of m^6^A peaks across sugarcane chromosomes. (**f**) The canonical RRACH and DRACH conserved sequence motifs of m^6^A peak regions in the CK sample.

**Figure 2 plants-12-03668-f002:**
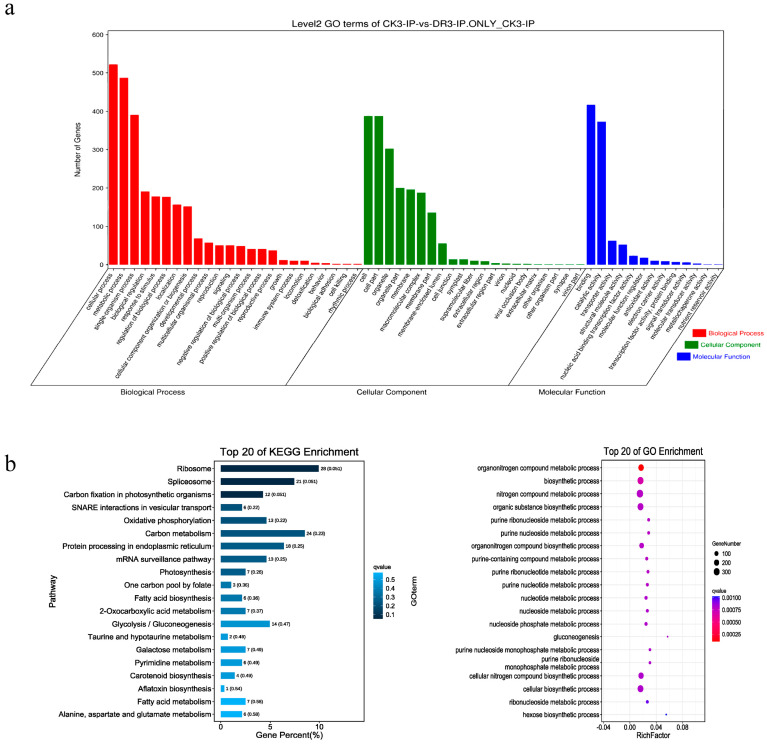
m^6^A methylation enrichment analysis. (**a**) GO enrichment of genes with m^6^A peaks. (**b**) KEGG enrichment of genes with m^6^A peaks. (**c**) GO enrichment of genes with m^6^A peaks in Peak I. (**d**) GO enrichment of genes with m^6^A peaks in Peak II. (**e**) GO enrichment of genes with m^6^A peaks in Peak III. (**f**) Examples of genes with m^6^A peaks at the stop codon. (**g**) Examples of genes with m^6^A peaks at the stop codon.

**Figure 3 plants-12-03668-f003:**
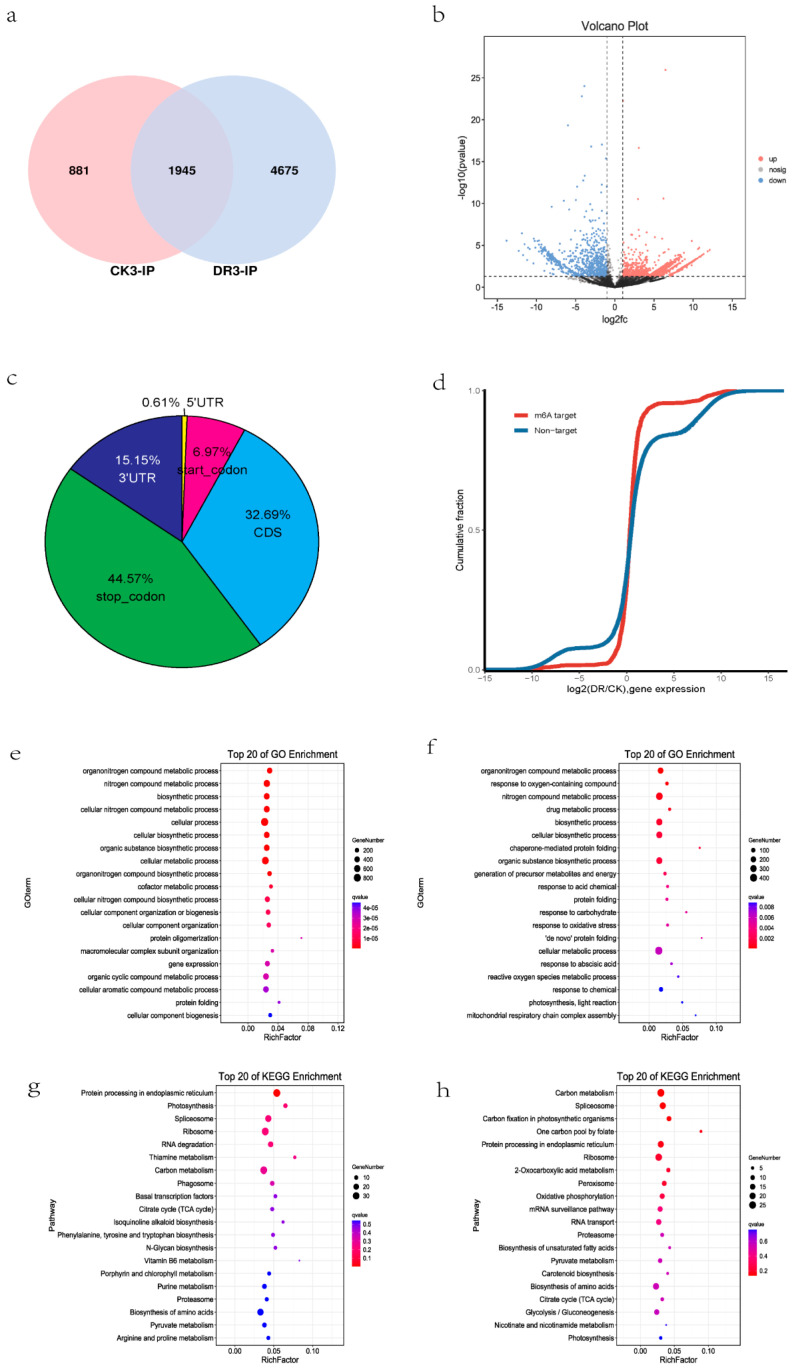
Overview of m^6^A methylation of sugarcane leaf under CK and DR treatment. (**a**) Venn diagram of m^6^A peaks between CK and DR. (**b**) Volcano analysis of differentially m^6^A methylation peaks. “nosig” is no significant changes. (**c**) Distribution of differentially m^6^A-methylated peaks across the mRNAs. (**d**) Cumulative distribution of mRNA expression changes between CK and DR for m^6^A-modified genes and non-target genes. (**e**) GO enrichment terms for the upregulated genes with DMPs between CK and DR. (**f**) GO enrichment terms for the downregulated genes with DMPs between CK and DR. (**g**) KEGG enrichment pathways of the upregulated genes with DMPs between CK and DR. (**h**) KEGG enrichment pathways of the downregulated genes with DMPs between CK and DR.

**Figure 4 plants-12-03668-f004:**
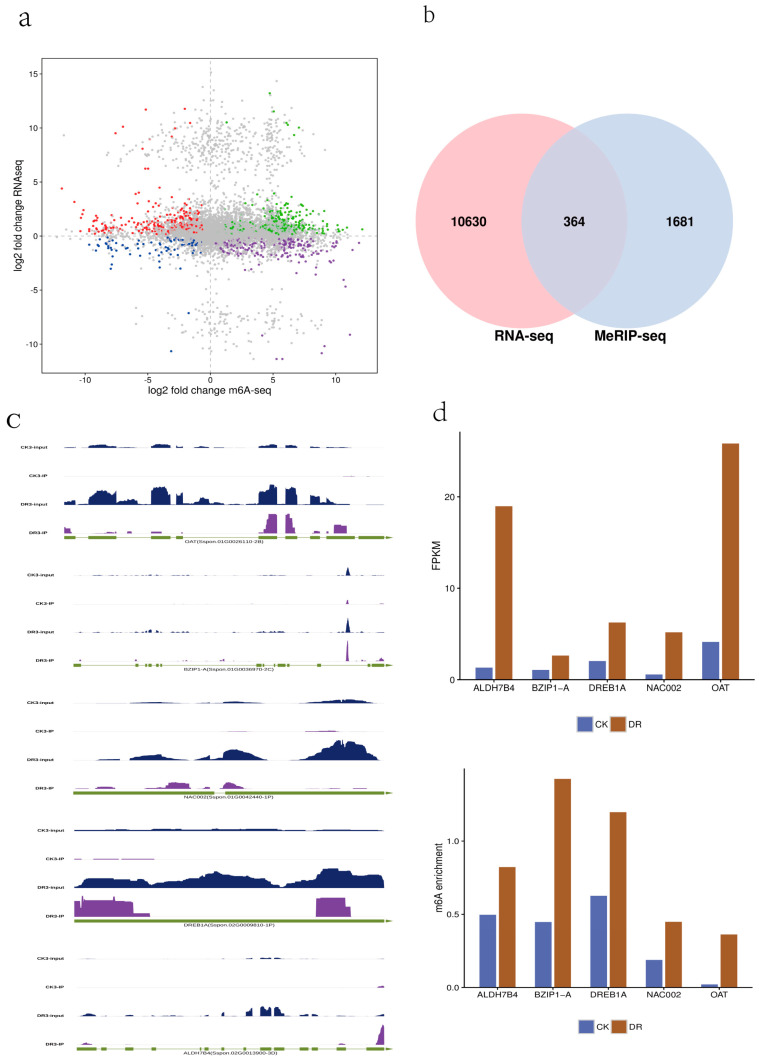
Analysis of DMP-related gene and DEGs between sugarcane leaf under CK and DRtreatment. (**a**) Venn map of DMP-related genes and DEGs. (**b**) Volcano map of DMP-related genes and DEGs. (**c**) Examples of DEGs with DMPs at the stop codon. (**d**) Expression level of DEGs with DMPs at the entire gene functional element.

**Figure 5 plants-12-03668-f005:**
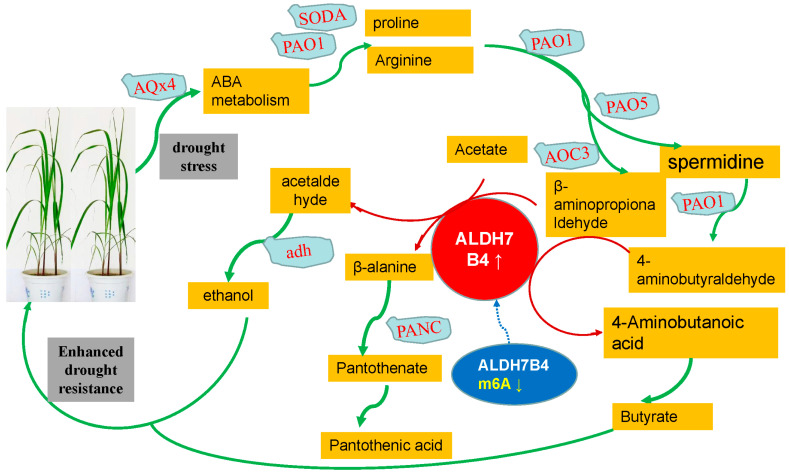
The action pattern of aldehyde dehydrogenase-related gene ALDH7B4 under drought stress.

**Figure 6 plants-12-03668-f006:**
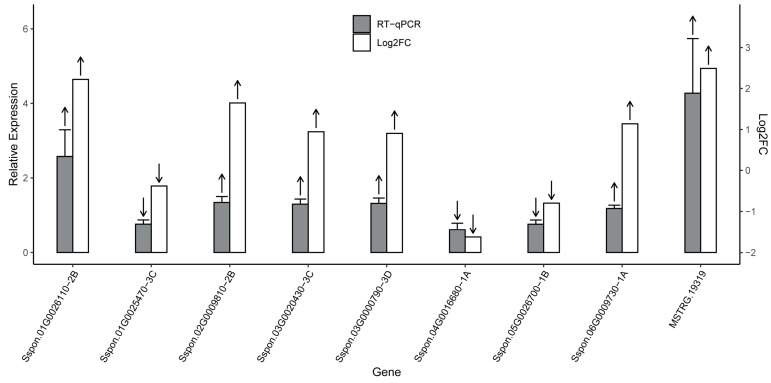
Quantitative real-time PCR analysis of selected differentially expressed genes.

**Table 1 plants-12-03668-t001:** The m^6^A level and methylation-related gene expression of 12 drought-resistant genes under drought stress.

ID	Gene_ID	Gene Name	DMG	DEG	Reference
Sspon.01G0036970-2C	AGT17406.1	bZIP	UP	UP	[22]
Sspon.01G0042440-1P	AGU13503.1	NAC	DOWN	UP	[23]
Sspon.01G0052010-1P	XP_002465468.1	HSP70	DOWN	UP	[24]
Sspon.02G0013900-1A	AMS36872.1	ALDH7	DOWN	UP	[21,25]
Sspon.04G0016680-1T	XP_002453419.1	AVP	DOWN	DOWN	[26]
Sspon.06G0009730-1A	XP_021321469.1	GLYI-11	DOWN	UP	[27]
Sspon.07G0010240-3C	OEL18927.1	WRKY	UP	UP	[28]
Sspon.07G0024880-3D	AFO59568.1	SODA	DOWN	UP	[29]
Sspon.08G0024020-1B	XP_002436386.1	DREB	DOWN	UP	[30,31]
Sspon.01G0026110-2B	XP_002464174.1	OAT	UP	UP	[18]
Sspon.07G0003250-3D	XP_002441446.2	PP2C	DOWN	UP	[32]
Sspon.07G0007070-2B	XP_021304140.1	cFBP	DOWN	UP	[33]

## Data Availability

The data presented in this study are available within the article and its Appendix A.

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
