# Peer review of "Coordination of m6A mRNA Methylation and Gene Transcriptome in Sugarcane Response to Drought Stress"

_plants, 2023, doi:10.3390/plants12213668_

Round 1
Reviewer 1 Report
Title: Coordination of m6A mRNA methylation and gene transcriptome in sugarcane response to drought stress
Sugarcane is primarily used for commercial sugar and biofuel production. Being a water-demanding crop, water scarcity or drought is the major environmental stress affecting sugarcane productivity. Though genetic and transcriptome studies have been conducted for different stresses, only few studies are available on epigenetic variation/modification on sugarcane for stresses more particularly on drought stress.
This study established the role of N6-methyladenosine (m6A) methylation of mRNA that involved in biological processes essential for plant growth using methylated RNA immunoprecipitation sequencing (MeRIP-seq) by constructing the m6A map of sugarcane. Using bioinformatics tools, this study proved the m6A sites of sugarcane transcriptome were significantly enriched around the stop codon and within 3ʹ-untranslated regions (3ʹUTR) and the m6A modification genes are associated with metabolic biosynthesis. Further, studied the m6A modification influence on drought-tolerance transcript mRNA differential expression of drought responsive genes. These findings provide a basis and resource for sugarcane RNA epigenetic studies and further increase our knowledge of the functions of m6A modifications in RNA under abiotic stress. As scarce information available on epigenetic studies on sugarcane stresses, this study adds a new dimension in this line for improved production under stressed environment.
However, I have some suggestions to improve the presentation of this manuscript.
Abstract: Concise covering a background and methodology performed and significant results obtained in this study.
Line 88, stop codon (60.30%),
Line 89, and the start codon (2.76???%)
Line 94, Figure 1e – not clearly visible. High resolution photo may be included.
Line 131, which plays a vital role in
Line 138, Not clearly visible. This figure may be presented in two parts to increase clarity and visibility to readers.
Line 166, Alignment of line and Figure e,f,g & h not clearly visible
Line 212, Alignment of line and Figure 4- X & Y -axis values/description are not clearly visible
Line 253, Alignment of line
Line 257-258, Six upregulated and three downregulated genes involved in plant biotic metabolism, photosynthesis, and defense responses were selected.
Line 267, Alignment of line
Results: Very comprehensive and detailed the results obtained in the study.
Discussion: Well discussed. Following recent references may appropriately be included as these works are more relevant to the study on drought stress.
Narayan Ashwin J, M.V., Gauri Nerkar, Chakravarthi M, Dharshini S, Subramonian N, Premachandran MN, Valarmathi R, Arun Kumar R, Gomathi R, Krisha Surendar K, Hemaprabha G, Appunu C. Transgenic sugarcane with higher levels of BRK1 showed improved drought tolerance. Plant Cell Rep. 2023, 42, 1611-1628.
Li, J., Phan, T.T., Li, Y.R., Xing, Y.X. and Yang, L.T., 2018. Isolation, transformation and overexpression of sugarcane SoP5CS gene for drought tolerance improvement. Sugar Tech, 20, pp.464-473.
Anunanthini, P., Manoj, V.M., Padmanabhan, T.S., Dhivya, S., Narayan, J.A., Appunu, C. and Sathishkumar, R., 2019. In silico characterisation and functional validation of chilling tolerant divergence 1 (COLD1) gene in monocots during abiotic stress. Functional Plant Biology, 46(6), pp.524-532.
Ramiro, D.A., Melotto‐Passarin, D.M., Barbosa, M.D.A., Santos, F.D., Gomez, S.G.P., Massola Junior, N.S., Lam, E. and Carrer, H., 2016. Expression of Arabidopsis Bax Inhibitor‐1 in transgenic sugarcane confers drought tolerance. Plant Biotechnology Journal, 14(9), pp.1826-1837.
Clarancia, P.S., Valarmathi, R., Suresha, G. S., Hemaprabha, G., Appunu, C., 2021. Isolation and characterization of drought responsive Aldehyde dehydrogenase (ALDH) gene from drought tolerant wild relative of sugarcane, Erianthus arundinaceus. Journal of Sugarcane Research, 11(2), pp.180-190. https://doi.org/10.37580/JSR.2021.1.11.56-65https://doi.org/10.1071/FP18189
Materials and methods: Given in detail and easy to follow by the readers.
Conclusion: Appropriate based on the results
Author Response
- Line 88, stop codon (60.30%),
Response: Thanks for your comments. We have changed 60.3 to 60.30.
- Line 89, and the start codon (2.76???%)
Response: Thanks for your comments. We have added 2.76.
- Line 94, Figure 1e – not clearly visible. High resolution photo may be included.
Response: Thanks for your comments. The attachment contains high resolution photo.
- Line 131, which plays a vital role in
Response: Thanks for your comments. We have changed the expression. Please see line 132-134.
- Line 138, Not clearly visible. This figure may be presented in two parts to increase clarity and visibility to readers.
Response: Thanks for your comments. We divided figure2 into three parts,like this: figure2-1、figure2-2 and figure2-3, Please see line 140-142.
- Line 166, Alignment of line and Figure e,f,g & h not clearly visible
Response: Thanks for your comments. We alignment of line and figure e, f, g & h. Please see line 175.
- Line 212, Alignment of line and Figure 4- X & Y -axis values/description are not clearly visible
Response: Thanks for your comments. We adjusted the picture. Please see line 222.
- Line 253, Alignment of line
Response: Thanks for your comments. We adjusted the lines of the table. Please see line 262.
- Line 257-258, Six upregulated and three downregulated genes involved in plant biotic metabolism, photosynthesis, and defense responses were selected.
Response: Thanks for your comments. The syntax expression has been modified as suggested. Please see line 268-269.
- Line 267, Alignment of line
Response: Thanks for your comments. We have adjusted the format. Please see line 278.
- Discussion: Well discussed. Following recent references may appropriately be included as these works are more relevant to the study on drought stress.
Response: Thanks for your comments. We have added the references as suggested. Please see Line 340.
Reviewer 2 Report
The research presented by Wei et al entitled, “Coordination of m6A mRNA methylation and gene transcriptome in sugarcane response to drought stress” offers a comprehensive and detailed investigation into m6A methylation in sugarcane leaves under drought stress. The findings are scientifically robust and hold significant implications for understanding the molecular mechanisms underlying plant responses to environmental challenges. Using an m6A-targeted antibody, coupled with MeRIP-seq, enabled the generation of a high-resolution m6A map. m6A peaks in sugarcane and mammals were found to be similar, reinstating the conservation of this modification across species. The distribution patterns of m6A peaks, especially around 3' UTRs and stop codons, mirror those seen in other plants, underscoring the significance of this modification in gene regulation. Furthermore, the study delves into the potential functional roles of m6A modifications. The enrichment analysis of genes modified by m6A under drought stress illuminates their involvement in metabolic biosynthesis, shedding light on the broader impact of m6A methylation on plant development.
The differential analysis between control and drought-stressed sugarcane leaves reveals a notable increase in m6A methylation under drought stress. The correlation between differentially methylated peaks (DMPs) and differentially expressed genes (DEGs) provides compelling evidence of the regulatory role of m6A in gene expression. The identified key drought-resistant genes, such as ALDH7, exemplify how changes in methylation status can influence mRNA abundance, potentially enhancing plant resilience to drought stress.
Moreover, the study highlights the intricate relationship between m6A methylation and key metabolic pathways, including proline biosynthesis. The downregulation of m6A levels in the aldehyde dehydrogenase gene (ALDH7) coupled with increased mRNA expression showcases a potential mechanism through which plants bolster their drought resistance. The verification of differential gene expression through RT-PCR further solidifies the credibility of the findings and corroborates the m6A-seq quantification results.
In summary, this study offers a valuable contribution to understanding m6A methylation in sugarcane under drought stress. The meticulous methodology, comprehensive analyses, and insightful interpretations make this research suitable for publication. It holds promise for informing strategies to enhance crop resilience in changing environmental conditions.
I have a few minor concerns that can be easily addressed by the authors:
1. In line 89, % of start codon is missing.
2. In figure 1f, the y-axis is labelled 'information content' and can be rectified.
3. In figure 3b, the grey dots are labelled 'nosig'. I believe the authors mean 'no significant changes', if so, kindly mention it in the figure legend.
4. Figure 4c can be rearranged so that it does not overlap with figure 4a above.
5. In figure 6, please mention what the arrows stand for.
Author Response
- In line 89, % of start codon is missing.
Response: Thanks for your comments. We have added the 2.76. Please see line 89.
- In figure 1f, the y-axis is labelled 'information content' and can be rectified.
Response: Thanks for your comments. We changed the “information content” into “Bits”. Please see line 109.
- In figure 3b, the grey dots are labelled 'nosig'. I believe the authors mean 'no significant changes', if so, kindly mention it in the figure legend.
Response: Thanks for your comments. We add the “nosig” in the paper. Please see line157-161.
- Figure 4c can be rearranged so that it does not overlap with figure 4a above.
Response: Thanks for your comments. We have reformatted figure 4. Please see line 222.
- In figure 6, please mention what the arrows stand for.
Response: Thanks for your comments. The upward arrow indicates the up-regulated gene, and the downward arrow indicates the down-regulated gene, which we have marked in the document.